# Classification with a Network of Partially Informative Agents: Enabling Wise Crowds from Individually Myopic Classifiers

## Abstract

We consider the problem of classification with a (peer-to-peer) network of heterogeneous and partially informative agents, each receiving local data generated by an underlying true class, and equipped with a classifier that can only distinguish between a subset of the entire set of classes. We propose an iterative algorithm that uses the posterior probabilities of the local classifier and recursively updates each agent's local belief on all the possible classes, based on its local signals and belief information from its neighbors. We then adopt a novel distributed min-rule to update each agent's global belief and enable learning of the true class for all agents. We show that under certain assumptions, the beliefs on the true class converge to one asymptotically almost surely. We provide the asymptotic convergence rate, and demonstrate the performance of our algorithm through simulation with image data and experimented with random forest classifiers and MobileNet.

## 1 INTRODUCTION

With the improvement of computation and communication technologies comes a new paradigm of problem solving with distributed intelligent agents. For example, Internet of Things (IoT) Li et al. (2015) retrieve information from sensors and process it at a centralized server. Edge Computing Shi et al. (2016) allows data produced by IoT devices to be processed locally, taking advantage of distributed computing to reduce communication latency to the hub and to improve data security.

Classification over a distributed network of intelligent sensor agents is essential to many applications, such as object or image identification González-Briones et al. (2018), anti-spam networks Luo et al. (2007), object tracking, etc. In these applications, the agents receive and process signals in real time. However, the distributed agents can be restricted by their computation and communication capabilities. Thus, given the private signals that each sensor collects, it is important to design efficient online (i.e., real-time) algorithms for the information of each agent to be integrated over time and propagated through the network.

In addition to communication and computation challenges, distributed agents often possess partial knowledge and must make decisions within constraints. For example, in third-generation surveillance systems Valera & Velastin (2005) with a large number of monitoring points, camera sensors are limited by their field of view Patricio et al. (2006). Environmental and industrial monitoring involve agents with diverse sensor types collecting various data Valverde et al. (2011). Multi-agent object recognition, first introduced by Yanai & Deguchi (1998), utilizes low-cost agents identifying only a single class, to achieve wide-range recognition by increasing the number of agents and including human agents YeeWai et al. (2020). In activity recognition scenarios, various sensors, each limited in its ability to obtain full information, e.g., accelerometer, gyroscope, and magnetometer in mobile phones, are employed to classify human activities, and various vehicle sensors are utilized to classify vehicles Smith et al. (2017).

Modern machine learning models, such as deep neural networks with millions or billions of parameters (e.g., Simonyan & Zisserman (2014), He et al. (2016)), can achieve high performance in classification tasks for a large number of classes. However, they are very expensive to be utilized in an online setting, requiring a large amount of training and inference resources. The constraints of computational resources make the learning and the implementation of such machine learning models very challenging on distributed networks of sensors,

such as robotic networks, wireless sensor networks, or IoT. Additionally, some models are pre-trained and developed, but might only be adequate for a subset of the given classification tasks. With the limitations described above, we ask: Can we utilize a network of partially informative machine learning models, each specialized in distinguishing a small number of classes, to achieve real-time identification of the true class in the entire network?

**Contributions**

In this paper, we aim to address the online distributed classification problem where heterogeneous agents are limited in computational resources and partially informative, i.e., their local classifiers can only distinguish between a subset of classes but provide no information on the classes outside that subset.

We propose a local update rule that can be applied with any local classifier generating posterior probabilities on the set of possible classes. Our local update rule incorporates data arriving in real-time to enhance the stability of estimation performance and robustness to noise.

We leverage the latest concepts from non-Bayesian distributed hypothesis testing and adopt a novel min-based global update rule. Utilizing only the partially informative belief vectors, each agent can asymptotically identify the true class given observations over time. We analyze the asymptotic convergence rate of the beliefs of agents, demonstrating that they can collectively reject the false classes exponentially fast. Finally, we show by simulation that our proposed approach outperforms other aggregation rules such as average and maximum.

**Related Literature**

**Distributed non-Bayesian social learning**

Distributed non-Bayesian learning considers the problem of identifying the true class over a network of distributed (peer-to-peer) agents. Each agent maintains a belief vector over a set of hypotheses and updates the beliefs sequentially. An agent's belief update is considered non-Bayesian as it treats the beliefs generated through interacting with neighbors as Bayesian priors rather than conditioning on all information available Jadbabaie et al. (2012).

Prior works assume each agent has exact and complete knowledge of the local likelihood functions of all classes, known as the private signal structure (e.g., Jadbabaie et al. (2012); Nedić et al. (2017); Lalitha et al. (2018); Mitra et al. (2021)). Other works attempt to estimate these likelihood functions (e.g., Hare et al. (2021)). These assumptions necessitate domain knowledge of the generative mechanism and all sensor characteristics and can cause model misspecifications or introduce additional uncertainties. Distributed non-Bayesian learning then considers scenarios where subsets of classes are observationally equivalent (i.e., the conditional likelihood distributions of given signals are identical) at each agent, which necessitates cooperative decision-making to identify the true class.

Our problem setup is fundamentally different from the existing work in distributed non-Bayesian social learning. Instead of relying on complete knowledge of the likelihood function and signal structure, our work leverages advancements in machine learning and directly utilizes the posterior probability provided by classifiers to identify the true class. These classifiers can include both discriminative models (such as random forests and neural networks, which often outperform generative models) and generative models that utilize likelihood functions (such as Naive Bayes). We assume that each agent is partially informative, i.e., each can provide information and distinguish between only a subset of classes while providing no information on the others. We expand upon these critical differences in more detail in Section 3.

**Ensemble learning**

Ensemble learning Zhang & Ma (2012); Sagi & Rokach (2018) is the generation and combination of multiple base learners to solve machine learning tasks. Some common output fusion methods are weighting (e.g., voting, Bayesian combination, and linear combination), and meta-learning methods (e.g., mixture of experts and stacking).

Weighting combines model outputs by assigning weights to each base model and is used in popular models such as Boosted Trees and Random Forests Sagi & Rokach (2018). However, the weighting approach is most suitable when the performance of the base models is comparable. It is not suitable for the scenario that we are considering, where each agent only provides information on a subset of classes.

Meta-learning models contain more than one learning phase. Mixture of experts is a learning technique where a task is divided into multiple sub-tasks and one or a few experts are selected from an ensemble of learners to solve specific sub-tasks Yuksel et al. (2012). Mixture of experts typically requires a trained gating mechanism or a centralized manager to select which expert to use given the input. Similarly, stacking trains an ensemble of models and learns a model to combine the predictions. In this work, we propose a local update rule to incorporate predictions over time, and utilize a global update rule that requires no training and can achieve learning even when the agents are fully distributed. In a fully connected network, our problem essentially simplifies to an instance of ensemble learning. In this case, our proposed update rule and min-rule play significant roles in ensuring robust performance, where the update rule and the min-rule contribute to effective real-time learning and the rejection of false classes to enhance overall classification accuracy.

### Distributed classification

In their work of distributed classification, Kotecha et al. (2005); Predd et al. (2006) consider that information from all sensors is gathered at a fusion center. In Kokiopoulou & Frossard (2010), a fully distributed consensus-based approach is proposed considering multi-observations of the same object and proposes a non-parametric approach; other works, such as Forero et al. (2010), cast the distributed learning problem as a set of decentralized convex optimization subproblems but are classifier specific and cannot be used for complex models such as deep neural networks or heterogeneous models.

Distributed learning is a critical research field, especially in the context of multi-agent systems where a group of agents collaborates to achieve a shared goal. Our problem uniquely considers a fully distributed network of agents aiming to discern the true class of the world using their local partially informative classifier of any type, through communication and cooperation with neighboring agents.

## 2 PROBLEM FORMULATION

### 2.1 Observation and Agent Model

We begin by defining $\Theta = \{\theta_1, \theta_2, \ldots, \theta_m\}$ as the set of $m \in \mathbb{N}$ possible classes of the world, where each $\theta_i \in \Theta$ is called a class. At each time-step $t \in \mathbb{N}$, $n \in \mathbb{N}$ data points $\{x_{1,t}, \ldots, x_{n,t}\}$ are generated from an unknown true class $\theta^* \in \Theta$. Each data point $x_{i,t} \in \mathcal{X} \subset \mathbb{R}^d$ is an input vector within a $d$-dimensional $(d \in \mathbb{N})$ finite input space. We assume that these data points are identical and independently distributed across time. However, at a given time step, the data points may be correlated.

Consider a group of $n \in \mathbb{N}$ agents (e.g., robots, sensors, or people). At each time step $t$, each agent $i$ observes a private data sample $x_{i,t} \in \mathcal{X}$. Each agent knows the set of possible classes $\Theta$ and is equipped with a locally pre-trained approximated mapping function (classifier) $f_i : \mathcal{X} \to \mathcal{P}_i$, where $\mathcal{P}_i \subset \mathbb{R}^{|\Theta_i|}$ is the probability measure such that $\sum_{\theta \in \Theta_i} p_i(\theta|x) = 1$. This classifier transforms the input $x \in \mathcal{X}$ into posterior probabilities $\{p_i(\theta|x) \in \mathcal{P}_i : \forall \theta \in \Theta_i\}$, which represents the probability of class $\theta \in \Theta_i$ given the observed input $x$. Due to limitations (such as during training), each agent can only distinguish a subset of the classes $\Theta_i \subseteq \Theta$, while providing no information on the other classes $\Theta \setminus \Theta_i$. An agent is considered partially informative if it can only distinguish a proper subset of all possible classes.

**Definition 2.1.** (Partially Informative) An agent $i$, equipped with classifier $f_i$, is partially informative if $\Theta_i \subset \Theta$.

These agents communicate via an undirected graph $\mathcal{G}_a = (\mathcal{V}_a, \mathcal{E}_a)$, where $\mathcal{V}_a = \{1, 2, \ldots, n\}$ is the set of vertices representing the agents and $\mathcal{E}_a \subseteq \mathcal{V}_a \times \mathcal{V}_a$ is the set of edges. An edge $(i, j) \in \mathcal{E}_a$ indicates that agent $i$ and $j$ can communicate with each other. The neighbors of agent $i \in \mathcal{V}_a$, including agent $i$ itself, are

represented by the set $\mathcal{N}_i = \{j \in \mathcal{V}_a : (i,j) \in \mathcal{E}_a\} \cup \{i\}$, and is termed the inclusive neighborhood of agent $i$. In this work, we assume the communication graph $\mathcal{G}_a$ is connected and time-invariant. [1]

Our objective in this work is to design distributed learning rules that allow each agent $i \in \mathcal{V}$, equipped with a partially informative classifier $f_i$ to identify the true class $\theta^*$ of the world asymptotically almost surely by communicating and collaborating with its neighbors.

## 2.2 Quality of Classifiers

Considering the capabilities of each agent and its corresponding local classifier, we introduce the following notations to describe the distinct roles of agents.

**Definition 2.2.** The **discriminative score** used to evaluate the capability of a classifier between two classes $\theta_p \in \Theta_i$ and $\theta_q \in \Theta_i$ is given by

$$D_i(\theta_p, \theta_q) \triangleq \sum_{x \in \mathcal{X}} p_i(x|\theta_p) \log \frac{p_i(\theta_p|x)/p_i(\theta_p)}{p_i(\theta_q|x)/p_i(\theta_q)}. \tag{1}$$

Here, $p_i(x|\theta_p)$ is the likelihood of seeing data $x$ given that class $\theta_p \in \Theta_i$ is true, and $\forall \theta \in \Theta_i$, $p_i(\theta)$ is the prior probability of class $\theta$ being true without any conditions. To ensure the definition is valid, we assume that the posterior and prior probability of each agent is non-zero, i.e., $p_i(\theta|x) > 0, p_i(\theta) > 0, \forall \theta \in \Theta$ and $\forall x \in \mathcal{X}$. This discriminative score can be interpreted as the expected information per sample in favor of $\theta_p$ over $\theta_q$ when $\theta_p$ is true.

Using the discriminative score, we define a source agent.

**Definition 2.3.** An agent $i$ is said to be a **source agent** for a pair of distinct classes $\theta_p, \theta_q \in \Theta$ if the discriminative score $D_i(\theta_p, \theta_q) > 0$. The set of source agents for $\theta_p, \theta_q$ is denoted as $\mathcal{S}(\theta_p, \theta_q)$. An agent $i$ is said to be a source agent for a set of classes $\Theta_i \subseteq \Theta$ if agent $i$ is a source agent for all pairs of $\theta_p, \theta_q \in \Theta_i$.

A source agent for a pair $\theta_p, \theta_q \in \Theta$ can distinguish between the pair of classes $\theta_p, \theta_q$ using its private signals $x$ and its posterior $p_i(\theta|x)$ obtained through its mapping function, i.e., classifier $f_i$. We assume each agent can distinguish between the classes in $\Theta_i$, as follows.

**Assumption 2.4.** Each agent $i \in \mathcal{V}_a$ is a source agent of classes $\Theta_i \subseteq \Theta$ and $\Theta_i \neq \emptyset$.

The agents can obtain the approximated mapping function $f_i$ through discriminative methods or generative methods. Classifiers that do not directly support probability predictions, such as Support Vector Machine and k-Nearest Neighbors, can use calibration methods such as Niculescu-Mizil & Caruana (2005); Platt et al. (1999) to obtain the probability for each respective label.

In order for all agents to identify the true class, for each pair of classes, there must exist at least one agent in the network who can distinguish that pair. In our distributed and partially informative scenario, we consider the following conditions.

**Assumption 2.5.** (Global Identifiability)[Mitra et al. (2021)] For each pair $\theta_p, \theta_q \in \Theta$ such that $\theta_p \neq \theta_q$, the set $\mathcal{S}(\theta_p, \theta_q)$ of agents that can distinguish between the pair $\theta_p, \theta_q$ is non-empty.

The global identifiability assumption is necessary under independent signals and is standard in related social learning literature (e.g., Jadbabaie et al. (2012), ensuring no class $\theta \neq \theta^*$ is observationally equivalent to $\theta^*$ for all agents in the network.

Additionally, we define the confusion score of agent $i$, which captures the following: given data generated by class $\theta^* \notin \Theta_i$, whether agent $i$ believes the data belongs to class $\theta_p \in \Theta_i$ or $\theta_q \in \Theta_i$.

**Definition 2.6.** The **confusion score** used to evaluate the capability of a classifier between two classes $\theta_p, \theta_q \in \Theta_i$ and $\theta^* \notin \Theta_i$ is given by

$$D_i^{\theta^*}(\theta_p, \theta_q) = \sum_{x \in \mathcal{X}} p_i(x|\theta^*) \log \frac{p_i(\theta_p|x)/p_i(\theta_p)}{p_i(\theta_q|x)/p_i(\theta_q)}. \tag{2}$$

---

[1] For broader assumptions regarding communication topology, we direct readers to Mitra et al. (2021).

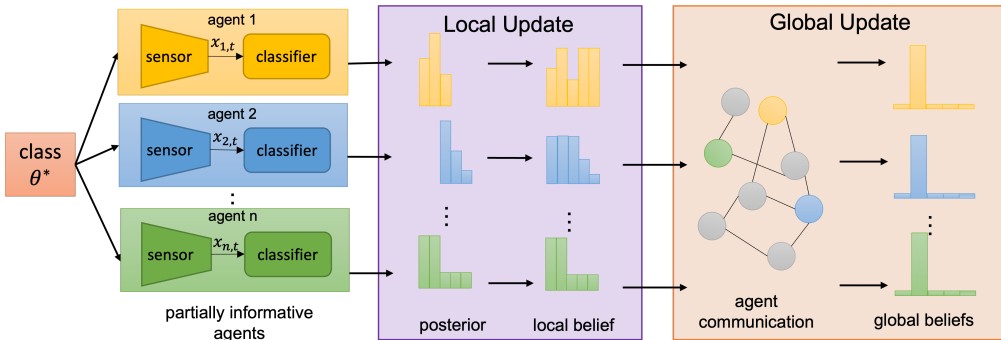

Figure 1: Demonstration of one time step of the proposed multi-agent classification algorithm.

If $D_i^{\theta^*}(\theta_p, \theta_q) > 0$, agent $i$ thinks $\theta_p$ is more likely to be the true class over $\theta_q$, and thus rejects $\theta_q$ from the set of possible candidate classes of being the true class. If $D_i^{\theta^*}(\theta_p, \theta_q) < 0$, agent $i$ rejects $\theta_p$ from the set of possible candidate classes of being the true class. If $D_i^{\theta^*}(\theta_p, \theta_q) = 0$, agent $i$ cannot reject either $\theta_p$ or $\theta_q$.

Note that the discriminative score and confusion score defined above are not required for our algorithm to obtain estimates, but provide information and performance guarantees on the models. To estimate the scores, one needs knowledge of the likelihood function $p_i(x|\theta)$ which is sometimes unknown in real-world applications. In practice, the likelihood function can be estimated via sampling while fixing the class.

Based on the above discussion, we define a support agent who can assist in rejecting false classes.

**Definition 2.7.** Consider an agent $i$ such that $\theta^* \notin \Theta_i$. The agent $i$ is said to be a **support agent** for a class $\theta \in \Theta_i$ if there exists some class $\hat{\theta} \in \Theta_i \setminus \{\theta\}$, such that the confusion score $D_i^{\theta^*}(\hat{\theta}, \theta) > 0$. We denote the set of support agents who can reject $\theta$ as $\mathcal{U}^{\theta^*}(\theta)$.

## 3 Proposed Learning Rules

In this section, we propose our local and global update rules. We include an overview of the update rules in Fig. 1. First, we propose the local update rule for each agent to update its local belief vectors $\pi_{i,t}$ for all $\theta \in \Theta$, and discuss how to address the issue with the partially informative classifier of each agent. Subsequently, we describe the communication and update rules for global belief vectors $\mu_{i,t}$ for all $\theta \in \Theta$ of each agent.

### 3.1 Local Update of Posterior

In this subsection, we first present the derivation leading to the local updates and subsequently discuss modifications that account for the partial information available to each agent.

Standard non-Bayesian social learning assumes that each agent has knowledge of its local likelihood functions $\{p_i(\cdot|\theta_k)\}_{k=1}^m$ (e.g., Mitra et al. (2021)). However, to obtain the likelihood functions, one has to make assumptions on the underlying distributions of the data. Here, we propose a method that directly uses the posterior probabilities from classifiers $f_i$.

We let $p_i(\theta|x_{i,t})$ be the posterior probability from the classifier of agent $i$, upon seeing private data $x_{i,t}$ at time step $t$. Note that since each agent $i$ is partially informative (i.e., $\Theta_i \subset \Theta$), the probability vector has dimension $|\Theta_i|$. We use the idea of adjusting a classifier to a new priori probability from Saerens et al. (2002) and recursively apply it to the outcomes of the classifier. Suppose agent $i$ receives a new data point $x_{i,t}$ at time-step $t$. According to Bayes' theorem, the likelihood of observing $x_{i,t}$ given class $\theta$ is characterized by

$$p_i(x_{i,t}|\theta) = \frac{p_i(\theta|x_{i,t})p_i(x_{i,t})}{p_i(\theta)}, \forall \theta \in \Theta_i, \tag{3}$$

where we directly obtain the posterior probability $p_i(\theta|x_{i,t})$ from the trained classifier of agent $i$, $p_i(\theta)$ is the probability of observing class $\theta$ without any conditions, and $p_i(x_{i,t})$ is the probability of observing data $x_{i,t}$ without any conditions.

Suppose we want to utilize prior knowledge and adjust the posterior probabilities $\tilde{p}_i(\theta|x_{i,t})$ with a new prior distribution $\tilde{p}_i(\theta)$. The adjusted posterior probability $\tilde{p}_i(\theta|x_{i,t})$ also obeys Bayes' theorem, with a new prior $\tilde{p}_i(\theta)$ and a new probability function $\tilde{p}_i(x_{i,t})$, i.e.,

$$\tilde{p}_i(\theta|x_{i,t}) = \frac{\tilde{p}_i(x_{i,t}|\theta)\tilde{p}_i(\theta)}{\tilde{p}_i(x_{i,t})}, \forall \theta \in \Theta_i. \tag{4}$$

We assume the likelihood distribution of the underlying generation mechanism does not change, i.e., $\tilde{p}_i(x_{i,t}|\theta) = p_i(x_{i,t}|\theta)$. By substituting Equation 3 into Equation 4 and defining $g_i(x_{i,t}) \triangleq p_i(x_{i,t})/\tilde{p}_i(x_{i,t})$, we obtain

$$\tilde{p}_i(\theta|x_{i,t}) = p_i(\theta|x_{i,t})g_i(x_{i,t})\frac{\tilde{p}_i(\theta)}{p_i(\theta)}, \forall \theta \in \Theta_i. \tag{5}$$

Since $\sum_{\theta \in \Theta_i} \tilde{p}_i(\theta|x_{i,t}) = 1$, we obtain $g_i(x_{i,t}) = 1/(\sum_{k=1}^{|\Theta_i|} p_i(\theta_k|x_{i,t})\frac{\tilde{p}_i(\theta_k)}{p_i(\theta_k)})$ and consequently, the adjusted posterior probability $\tilde{p}_i(\theta|x_{i,t})$ is

$$\tilde{p}_i(\theta|x_{i,t}) = \frac{p_i(\theta|x_{i,t})\frac{\tilde{p}_i(\theta)}{p_i(\theta)}}{\sum_{k=1}^{|\Theta_i|} p_i(\theta_k|x_{i,t})\frac{\tilde{p}_i(\theta_k)}{p_i(\theta_k)}}, \forall \theta \in \Theta_i. \tag{6}$$

With the above discussion on adjusting posterior probabilities given a new prior, we now describe how to update the local beliefs of each agent with partially informative classifiers. We define the local belief $\pi_{i,t}(\theta) \triangleq \tilde{p}_i(\theta|x_{i,1}, \ldots, x_{i,t}), \forall i \in \mathcal{V}_a$ as the posterior probability of $\theta \in \Theta_i$ after seeing all the data up to time $t$. At each iteration $t$, since we assume the data are i.i.d. given the class, each agent can incorporate its previously adjusted belief $\pi_{i,t-1}(\theta)$ as a prior and uses the posterior $p_i(\theta|x_{i,t})$, from its classifiers $f_i$, to update the belief $\pi_{i,t}$. Substituting $\tilde{p}_i(\theta) = \pi_{i,t-1}(\theta)$ into the numerator of Equation 6 and ignoring the effect of the denominator for now, we obtain the unnormalized local belief $\hat{\pi}_{i,t}$ as

$$\hat{\pi}_{i,t}(\theta) = \frac{p_i(\theta|x_{i,t})}{p_i(\theta)}\pi_{i,t-1}(\theta), \forall \theta \in \Theta_i. \tag{7}$$

Intuitively, if seeing the data point $x_{i,t}$ improves the posterior probability of class $\theta \in \Theta_i$, i.e., $p_i(\theta|x_{i,t}) > p_i(\theta)$, the local belief $\hat{\pi}_{i,t}(\theta)$ is increased, compared to its previous local belief $\pi_{i,t-1}(\theta)$. On the other hand, if receiving data $x_{i,t}$ suggests the true class is less likely to be $\theta \in \Theta_i$, i.e., $p_i(\theta|x_{i,t}) < p_i(\theta)$, the local belief $\hat{\pi}_{i,t}(\theta)$ is decreased, compared to its previous local belief $\pi_{i,t-1}(\theta)$.

Since agents are partially informative, in the case that $\theta \in \Theta \setminus \Theta_i$, we let

$$\hat{\pi}_{i,t}(\theta) = \max_{\theta_k \in \Theta_i} \hat{\pi}_i(\theta_k), \forall \theta \in \Theta \setminus \Theta_i. \tag{8}$$

The intuition of this modification is that agent $i$ will have a weaker belief on the classes that are less likely to be the true class, informed by its classifier; agent $i$ considers that the classes it cannot identify to be equally likely (i.e., with the same probabilities) of being the true class. This modification ensures that agent $i$ leaves open the possibility that the true class is one that it cannot identify based on its own classifier.

We apply normalization, such that $\sum_{\theta \in \Theta} \pi_{i,t}(\theta) = 1$, and obtain

$$\pi_{i,t}(\theta) = \frac{\hat{\pi}_{i,t}(\theta)}{\sum_{k=1}^m \hat{\pi}_{i,t}(\theta_k)}, \forall \theta \in \Theta. \tag{9}$$

The local update incorporates only the private observations and beliefs, without any network influence. We will show in Section 4 that for a source agent $i \in \mathcal{S}(\theta^*, \theta)$, $\pi_{i,t}(\theta) \to 0$ almost surely. However, without communication with neighbors, agents cannot identify $\theta^*$ just yet due to their partially informative classifiers. We will utilize the global update to propagate the beliefs such that every agent can identify the true class $\theta^*$.

### 3.2 Global Update

We define the global belief vector of an agent $i$ on class $\theta \in \Theta$ at time $t$ to be $\mu_{i,t}(\theta)$. Each agent determines the final estimation result using the global belief vector. At each time step $t$, once the local beliefs $\pi_{i,t}$ of all agents $i \in \mathcal{V}_a$ are updated, all agents perform a round of global update as in Mitra et al. (2021) to update their global belief vector $\mu_{i,t}$ as follows:

$$\mu_{i,t}(\theta) = \frac{\min\left\{\{\mu_{j,t-1}(\theta)\}_{j\in\mathcal{N}_i}, \pi_{i,t}(\theta)\right\}}{\sum_{k=1}^{m}\min\left\{\{\mu_{j,t-1}(\theta_k)\}_{j\in\mathcal{N}_i}, \pi_{i,t}(\theta_k)\right\}}, \forall\theta \in \Theta. \tag{10}$$

The intuition behind the update rule is that the agents go through a process of elimination and reject the classes with low beliefs. In the scenarios with partially informative agents, these agents do not rule out their locally unidentifiable classes. With the min-update, the source agent $i \in \mathcal{S}(\theta, \theta^*)$, who can distinguish a pair of classes $\theta, \theta^*$ with its local classifier, will contribute its information to the agent network and drive its neighbors' beliefs on the false class $\theta$ lower through the *min* operator. As proven in Mitra et al. (2021), the min-rule achieves faster asymptotic convergence rates than linear and log-linear updates.

For simplicity of analysis, we assume a connected communication network in this work. However, as demonstrated by Mitra et al. (2021), network-wide inference can be achieved as long as the source agent is reachable by other agents in the network. Furthermore, in the case of time-varying communication graphs, the algorithm remains effective when the union of the communication graph is jointly strongly connected. A similar analysis can be used to show that the update rule in our paper (leveraging posterior distributions directly instead of likelihoods as in Mitra et al. (2021)) will also work in time-varying networks (as long as the unions of the networks over bounded intervals of time are connected).

Although we adopt the global *min* update rule from Mitra et al. (2021), our problem formulation and theoretical performance guarantees differ significantly. Firstly, we bridge the gap between distributed classification and non-Bayesian social learning. Unlike prior works that require **precise** knowledge of the underlying generative mechanism and signal structure, we leverage posterior probabilities from classifiers. As illustrated in Fig. 1, while prior work directly uses outputs from sensors with known signal structures, our work directly leverages the outputs generated by any classifiers. Our work facilitates the application of both generative and discriminative classifiers, enhancing model flexibility and applicability. Secondly, our approach incorporates a unique input structure, where agents are partially informative, i.e., provide information for a subset of classes. In contrast, traditional social learning requires each agent to have **complete** likelihood functions for all possible classes, i.e., $\{p_i(\cdot|\theta_q)\}_{q=1}^m$. By adopting the proposed approach, we eliminate the extensive modeling efforts required to characterize sensor and signal structures, consequently reducing overall modeling and training requirements. Thirdly, we identify, define, and quantify the roles of support agents, a crucial aspect overlooked in prior work. These agents, while not able to directly distinguish the true class, contribute to rejecting false classes. In particular, we show that the performance is influenced by both source and support agents in the network, and offer performance guarantees that demonstrate a strict improvement over Mitra et al. (2021).

We include the posterior modification, local and global updates of agent $i \in \mathcal{V}_a$ in Algorithm 1.

## 4 Analysis of Convergence

In this section, we introduce assumptions and subsequently demonstrate the convergence of the local and global updates. The proofs of all theoretical results can be found in the appendix.

For the local and global update rules, if the initial belief is zero for any $\theta \in \Theta$, the beliefs of $\theta$ will remain zero in the subsequent updates. Thus, we assume the initial beliefs to be positive to eliminate this situation. In addition, to simplify the analysis, we let the assumption be satisfied by uniformly initializing the beliefs of all classes.

**Assumption 4.1.** Agent $i \in \mathcal{V}_a$ has positive initial beliefs and $\pi_{i,0}(\theta) = \frac{1}{|\Theta|}$ and $\mu_{i,0}(\theta) = \frac{1}{|\Theta|}, \forall\theta \in \Theta$.

The next results prove the correctness of the proposed local update.

---

**Algorithm 1** Classification for agent $i \in \mathcal{V}_a$

---

**Input:** At each time step $t$, each agent $i$ receives $x_{i,t} \in \mathbb{R}^p$ (unlabelled private observation)

**Initialization:** $\mu_{i,0}(\theta) = \frac{1}{|\Theta|}$, $\pi_{i,0}(\theta) = \frac{1}{|\Theta|}$, $\forall \theta \in \Theta$

At $t = 0$, transmit $\mu_{i,0}$ to neighbors and receive $\{\mu_{j,0}\}_{j \in \mathcal{N}_i}$

**for** $t \in \mathbb{N}$ **do**

    **for** every $\theta \in \Theta_i$ **do**

        Obtain posterior probabilities $p_i(\theta | x_{i,t})$

        $\hat{\pi}_{i,t}(\theta) = \frac{p_i(\theta | x_{i,t})}{p_i(\theta)} \pi_{i,t-1}(\theta)$

    **end for**

    **for** every $\theta \in \Theta \setminus \Theta_i$ **do**

        $\hat{\pi}_{i,t}(\theta) = \max_{\theta_k \in \Theta_i} \hat{\pi}_{i,t}(\theta_k)$

    **end for**

    **for** every $\theta \in \Theta$ **do**

        $\pi_{i,t}(\theta) = \frac{\hat{\pi}_{i,t}(\theta)}{\sum_{k=1}^{m} \hat{\pi}_{i,t}(\theta_k)}$

        $\mu_{i,t}(\theta) = \frac{\min\left\{\{\mu_{j,t-1}(\theta)\}_{j \in \mathcal{N}_i}, \pi_{i,t}(\theta)\right\}}{\sum_{k=1}^{m} \min\left\{\{\mu_{j,t-1}(\theta_k)\}_{j \in \mathcal{N}_i}, \pi_{i,t}(\theta_k)\right\}}$

    **end for**

    Transmit global belief vector $\mu_{i,t}$ to neighbors

    Receive neighbors' global belief vectors $\{\mu_{j,t}\}_{j \in \mathcal{N}_i}$

**end for**

---

**Theorem 4.2.** *Consider an agent $i \in \mathcal{S}(\theta, \theta^*)$. Suppose Assumptions 2.5 and 4.1 are satisfied, then, the update rules eqs. (7) to (9) ensure that for any $\theta \in \Theta \setminus \{\theta^*\}$ (i) $\pi_{i,t}(\theta) \to 0$ almost surely, (ii) $\lim_{t \to \infty} \pi_{i,t}(\theta^*) \triangleq \pi_{i,\infty}(\theta^*)$ exists almost surely and $\pi_{i,\infty}(\theta^*) \geq \pi_{i,0}(\theta^*) > 0$.*

Theorem 4.2 shows that for a source agent $i \in \mathcal{S}(\theta, \theta^*)$, its local belief on the false class $\theta$, $\pi_{i,t}(\theta) \to 0$ almost surely, via the proposed local updates. For a source agent, it can reject the false class asymptotically without the help of any other agents, while keeping the belief on the true class away from zero. In the next result, we show that for a support agent $i \in \mathcal{U}^{\theta^*}(\theta)$, its local belief on the false class $\theta \in \Theta_i$ goes to zero as well.

**Lemma 4.3.** *Consider an agent $i \in \mathcal{U}^{\theta^*}(\theta)$. Suppose Assumptions 2.5 and 4.1 are satisfied. Then the update rules eqs. (7) to (9) ensure that $\pi_{i,t}(\theta) \to 0$ almost surely.*

The above result states that with probability 1, a support agent will be able to rule out the false class $\theta$. Intuitively, given the input data, agent is able to reject the classes that are the furthest away in the feature space from the true class (the least likely to be a true class), even when the agent cannot distinguish the true class using its local classifier.

With the rejection of local false beliefs, next, we show that the global belief on the true class $\theta^*$ converges to 1 almost surely. To analyze the global convergence, recall that we assume the network is connected to simplify analysis. In general, we only require the source agents to have paths to other agents in the network.

**Theorem 4.4.** *Suppose Assumptions 2.5 and 4.1 are satisfied. Then, the update rules in Algorithm 1 lead to the learning of the true class for all agents, i.e., $\mu_{i,t}(\theta^*) \to 1$ almost surely for all $i \in \mathcal{V}_a$.*

The previous result shows that the true class $\theta^*$ will be identified by all agents in the network with probability 1. In the next theorem, we characterize the rate of rejection of any false class $\theta \in \Theta \setminus \{\theta^*\}$.

**Theorem 4.5.** *Suppose Assumptions 2.5 and 4.1 are satisfied. Then, for all $i \in \mathcal{V}_a$, for any false class $\theta \in \Theta \setminus \{\theta^*\}$, the update rules in Algorithm 1 guarantee the rejection of $\theta$ with the rate:*

$$\liminf_{t \to \infty} -\frac{\log \mu_{i,t}(\theta)}{t} \geq R_{v_\theta} \; a.s., \tag{11}$$

*where $R_{v_\theta} \triangleq \max_{i \in \mathcal{S}(\theta, \theta^*) \cup \mathcal{U}^{\theta^*}(\theta)} \{D_i(\theta^*, \theta), \max_{\hat{\theta} \in \Theta_i} D_i^{\theta^*}(\hat{\theta}, \theta)\}$ is the best rejection rate of false class $\theta$ and $v_\theta \in \arg\max_{i \in \mathcal{S}(\theta, \theta^*) \cup \mathcal{U}^{\theta^*}(\theta)} R_{v_\theta}$.*

With probability 1, each agent will be able to reject any false class $\theta$ exponentially fast, with a rate that is eventually lower-bounded by the best agent $v_\theta$ with the highest performance score $R_{v_\theta}$, either the best source agent $i \in \mathcal{S}(\theta^*, \theta)$ and its discriminative score $D_i(\theta^*, \theta)$ or the best support agent $i \in \mathcal{U}^{\theta^*}(\theta)$ and its confusion score $\max_{\hat{\theta} \in \Theta_i} D_i^{\theta^*}(\hat{\theta}, \theta)$ in the network. This lower bound is a strict improvement over Mitra et al. (2021), as both source agents and support agents contribute to the prediction convergence. Additionally, as stated in Mitra et al. (2021), the convergence rate is independent of network size and structure. In other words, the long-term (asymptotic) learning rate is independent on how the information is distributed among the agents.

## 5 Experiment and Simulation

### 5.1 Demonstrative Example

In this subsection, we provide an example with three classes and two agents to demonstrate our algorithm. The data distribution and agent classification boundaries can be found in Fig. 2.

Each agent $i$ has a classifier producing probabilities. Agent 0 employs a one-class classifier, represented by the circular decision boundary. If a data point $x$ sampled from $\theta_0$ is within the circle, agent 0 can correctly classify it, i.e., $p(\theta_0|x) = 0.8$, and if $x$ is sampled from any $\Theta \setminus \{\theta_0\}$, $p(\theta_0|x) = 0.2$. Agent 1 can distinguish classes 1 and 2 apart with its linear decision boundary. If a data point $x$ is on the left of the decision boundary, agent 1 identifies it as $\theta_1$, i.e., $p(\theta_1|x) = 0.8$ and $p(\theta_2|x) = 0.2$. Conversely, if the data is on the right of the decision boundary, agent 1 identifies it as $\theta_2$, i.e., $p(\theta_1|x) = 0.2$ and $p(\theta_2|x) = 0.8$.

Despite agent 0's inability to discriminate between classes 1 and 2, and agent 1's lack of information on class 0, the global identifiability assumption is still met. For every pair of classes, there is an agent capable of differentiating between them, i.e., $\mathcal{S}(\theta_0, \theta_1) = \{1\}$, $\mathcal{S}(\theta_0, \theta_2) = \{1\}$, and $\mathcal{S}(\theta_1, \theta_2) = \{2\}$.

With the classifier outlined above, each agent observes a data point drawn from $\theta_1$ (denoted by stars) and obtains posterior from its local classifier. The fictitious example is presented in Table 1. Using the posteriors, each agent applies the proposed local update rule, and subsequently, updates the global belief.

As shown in Table 1, the proposed local and global updates are more effective in identifying the true class with higher confidence. By rejecting false classes via the *min* operator, the proposed algorithm assigns the highest score for the true class. In contrast, the commonly studied averaging method results in a lower belief on the true class, while adopting the maximum leads to the agents' inability to identify the true class.

Achieving the performance demonstrated is impossible with methods proposed by Mitra et al. (2021) and most non-Bayesian social learning works. Prior works directly use outputs from sensors (see Fig. 1) with known signal structures and are based on the critical assumption that the local signal structures (characterized by likelihood functions) for all classes are known precisely. As depicted in Fig. 2, obtaining the likelihood functions for all classes of each agent is infeasible with only observation data at deployment. Even extensive training on the likelihood functions, such as Hare et al. (2021), requires knowledge of the family of distributions and introduces additional uncertainties. On the other hand, our proposed method allows for efficient classification, even when agents are equipped with simple discriminative classifiers. This signifies a substantial improvement in terms of practicality and efficiency for a broader application in real-world scenarios.

### 5.2 Dataset and Agent Network

In this simulation, we use the widely recognized CIFAR-10 image dataset Krizhevsky et al. (2009), containing 10 distinct classes with a total of 50,000 training images and 10,000 testing images.

We use an Erdos-Renyi graph with an edge generation probability of 0.5 for the communication topology of 9 agents as shown in Fig. 3. Each agent has access to all the training data of 4 classes (5000 images per class), as labeled, thus has capabilities of identifying the 4 classes, i.e., $|\Theta_i| = 4$. The subsets of identifiable classes $\Theta_i$ of each agent are selected such that the global identifiability condition is satisfied.

We train random forests as the classifiers for each agent. A random forest is an ensemble of decision trees Breiman (2001), and we obtain the posterior probability by averaging the probabilistic prediction of all

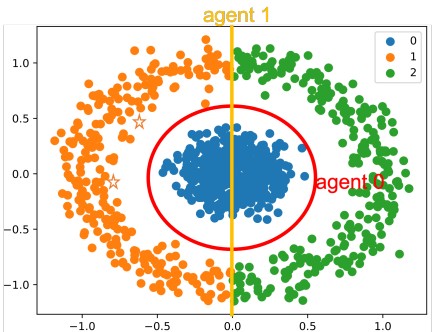

| | | $\theta_0$ | $\theta_1$ | $\theta_2$ |
|---|---|---|---|---|
| Posterior | agent 0 | 0.2 | | |
| | agent 1 | | 0.8 | 0.2 |
| Local belief | agent 0 | 0.12 | 0.44 | 0.44 |
| | agent 1 | 0.44 | 0.44 | 0.12 |
| Global belief | proposed | 0.18 | **0.64** | 0.18 |
| | averaging | 0.28 | 0.44 | 0.28 |
| | maximum | 0.44 | 0.44 | 0.44 |

Figure 2: Demonstrative example with 3 classes and 2 agents.

Table 1: Beliefs and comparisons of the demonstrative example.

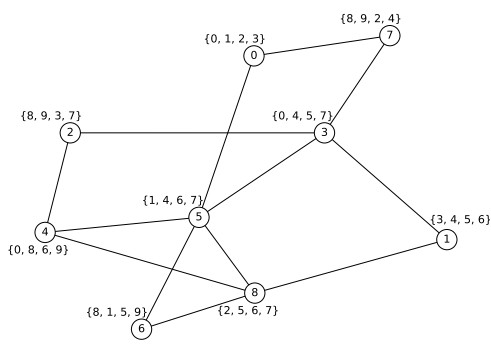

Figure 3: Agent communication topology and their distinguishable classes $\Theta_i$.

trees for each agent. In this demonstration, random forests are not the best classifier for the complex image classification task; however, we use them as weak classifiers to demonstrate the capabilities of our algorithm. For a centralized baseline, we train a random forest with 200 trees with all the training data (50,000 images) from CIFAR-10. For the distributed scenario, we independently train a random forest with 50 trees for each agent, using only the training data belonging to classes in $\Theta_i$.

Finally, to demonstrate the roles of source and support agents and the improvement in learning rate, we select a subset of agents and provide neural-network-based classifiers with higher accuracy (than random forest). We select MobileNet V3 Large Howard et al. (2019) for its relative light weight, good performance, and its model architecture design for mobile classification tasks. For fair comparisons (with the random forest set up), we tune MobileNets (weights pre-tained with ImageNet Howard et al. (2019)Russakovsky et al. (2015)) with all available CIFAR-10 training data, depending on the agent and its classes $\Theta_i$.

### 5.2.1 Local Update

In this simulation, we demonstrate the performance of the local update rule. We consider a centralized baseline random forest trained with data from all classes, i.e., $|\Theta_i| = 10$, as described in the previous subsection. The class with the highest probability is the estimated true class. The trained model achieved a training accuracy of 0.96 and a testing accuracy of 0.47. The confusion matrix of the model evaluated with testing data is shown in Fig. 15 (in the Appendix).

We select class "cat" (label 3) as the true class $\theta^*$ since it is the class with the lowest accuracy according to the confusion matrix. We independently and identically sample 150 cat images from the false negative (incorrectly classified as other classes) testing data and provide them sequentially to the classifier. The number of misclassifications of these images is shown in Fig. 16 (in the Appendix). The classifier cannot identify the true class $\theta^*$ for any data point.

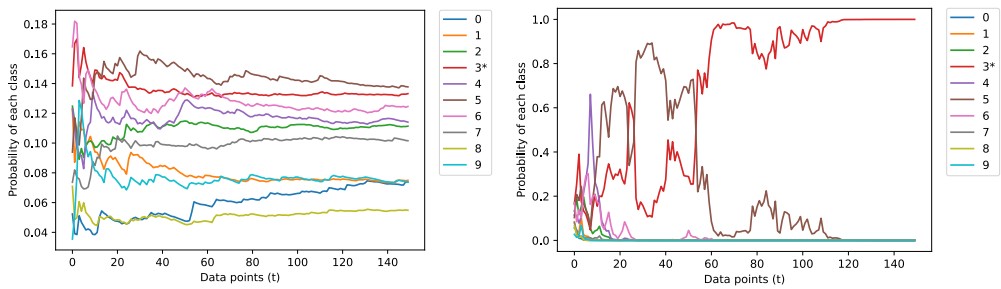

Figure 4: Local averaging                Figure 5: Proposed local update

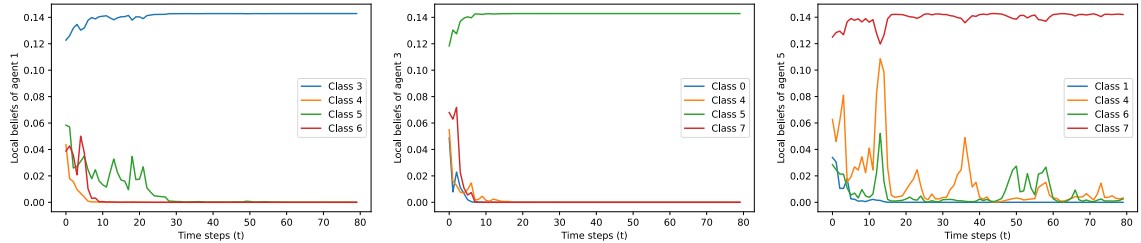

Figure 6: Local beliefs of agent   Figure 7: Local beliefs of agent   Figure 8: Local beliefs of agent
1.                                 3.                                 5.

We then consider simple averaging, where we average across all the posterior probabilities from the classifiers over all time steps. The results are shown in Fig. 4. However, the classifier cannot identify the correct class by averaging the probabilities. In Fig. 5, the posterior probabilities were updated according to Equation 7 and Equation 9, assuming uniform prior distributions. The classifier can identify the correct class at 60 time steps and the belief on the correct class $\pi(\theta^*)$ converges to 1.

### 5.2.2 Distributed Setting

We now consider the distributed setting with 9 agents as in Fig. 3, each independently trained with their 4 classes of training data and equipped with a random forest of 50 trees. At each time step, each agent is given an image that is identically and independently sampled from the testing data of class $\theta^*$ (cat).

In Fig. 6, Fig. 7, and Fig. 8, we include the local beliefs of agents for their respective observable classes $\Theta_i$. Each agent rejects all classes except one class that is the most likely to be the true class. However, because of the partial information of each agent, they cannot identify the true class just yet: they rely on the communication with neighbors to identify the true class.

In Fig. 9, Fig. 10, and Fig. 11, we include the beliefs of agents on the true (class 3) and false classes (selected to be class 5 and class 0), respectively. We include the beliefs of agent 1, who is a source agent in $\mathcal{S}(3,5)$, and support agent 3, 5, 7. We observe that the beliefs on the true class converge to 1 and the beliefs on the false class converge to 0 for all selected agents. Agent 1 was able to correctly identify the true class $\theta^*$ using its private and neighbors' information. Agents 3, 5, and 7, who have no capability of distinguishing between class 3 and class 5, were also able to identify the correct class by integrating neighbors' information. Note that the time steps scale of Fig. 11 is different than previous two figures, due to the fast rejection of class 0, thanks to support agent 3, who can reject class 0 efficiently. Following the proposed update rules, all agents were able to identify the correct class at about 10 time steps when $\mu_i(\theta^*) > \mu_i(\theta), \forall \theta \in \Theta \setminus \{\theta^*\}$.

### 5.2.3 Improvements in Learning Rate

We further examined the performance of the proposed algorithm, when selected agents are given powerful classifiers. More powerful classifiers will result in higher discriminative scores or higher confusion scores,

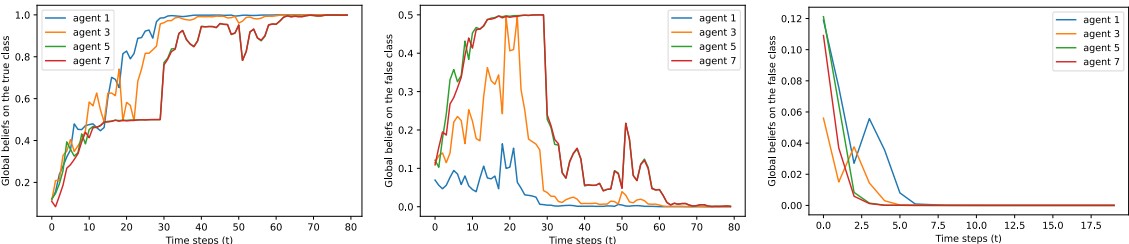

Figure 9: Global beliefs on true class 3.

Figure 10: Global beliefs on false class 5.

Figure 11: Global beliefs on false class 0.

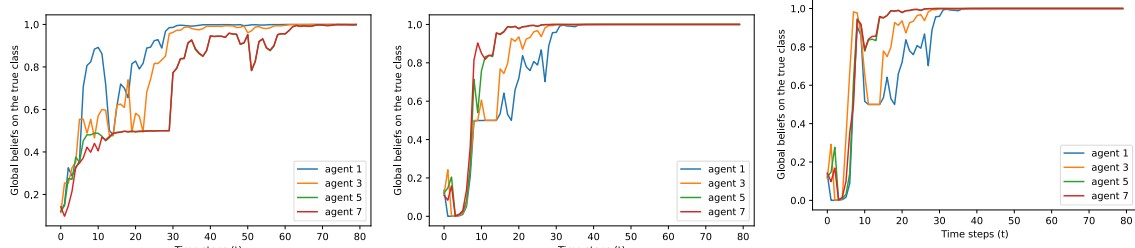

Figure 12: Global beliefs on true class with improved agent 7.

Figure 13: Global beliefs on true class with improved agent 1.

Figure 14: Global beliefs on true class with improved agent 1 and agent 7.

depending on the specific agents. In this experiment, with the application of distributed sensor networks in mind, we selected agent 1 and agent 7, and trained MobileNet V3 Large Howard et al. (2019) using all the training data in their respective classes $\Theta_i$. In Fig. 12, we only give agent 7 a trained neural network. However, the performance improvement is limited. This is due to the fact that agent 7 is not a source agent, nor can it distinguish the most confused classes, 3 and 5. In Fig. 13, we only give agent 1 a trained neural network. As shown in the figure, the performance of all selected agents are significantly improved, since agent 1 is a source agent $i \in \mathcal{S}(3, 5)$. Finally, in Fig. 14, we give agent 1 and agent 7 their respective trained neural networks. This results in an overall better transient behavior and asymptotic convergence.

## 6 Conclusions and Future Work

We proposed a distributed algorithm to solve the classification problem with a network of partially informative and heterogeneous agents. We based our algorithm on a simple recursive local update and a min-rule-based global update. We provided theoretical guarantees of the convergence and demonstrated the performance of our algorithm through simulation with image data and experimented with random forest classifiers and MobileNet.

For future work, we are looking into improving the performance of the proposed algorithm through training different agents with data of different feature spaces, such as images, sounds, temperatures, etc. We will also explore the use of calibration techniques to improve the transient behaviors of the proposed algorithm.

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
