# OpenReview forum: "Classification with a Network of Partially Informative Agents: Enabling Wise Crowds from Individually Myopic Classifiers"
_TMLR — Rejected by TMLR_

### Review · Reviewer_rDXw · 2023-03-16

**Summary Of Contributions:**

This paper designs an algorithm to for source identification in a distributed setting.  Each agent has access to local features that allow her to distinguish between a few classes. The authors design an algorithm to combine these local information in order to identify the correct source.

This paper seems to extend a previous paper Mitra et al (2021) to a decentralized case where agents are connected through a graph interaction.


**Audience:**

Yes

**Broader Impact Concerns:**

N/A.

**Claims And Evidence:**

No

**Requested Changes:**

To me, it is hard to judge the contribution of the paper because and I do not understand the motivation and not even the theoretical model itself. I think that section 2 should be completely rewrote. For instance: start by section 2.4 and then explain what are private signals, classes, etc. (reading the paper, it is not easy to guess that x is a private value). Also, many things are defined multiple times (ex: a source agent can discriminate, function f that becomes p,...), each time with a different sentence or notation. This makes the confusion even harder. Also:
- what is p_i(\theta) and why should this be known to agent i?
- what do the agents know exactly?

Similarly, Section 3 starts with a long discussion about (Equations (3)-(6)) before ending with an algorithm that seems much simpler. Where is the catch?

Some discussion about why this problem is interesting should be added.

A better connection with related work would be appreciated. This work seems to be a direct extension of Mitra et al. (2021)

I understand that there are space constraints but some element of proofs would be appreciated.



**Strengths And Weaknesses:**

Strengths:
- Results seem correct (if one assume that the model is). The appendix are generally well written.

Weaknesses:
- the problem formulation (Section 2) is not clear to me. In particular, I do not understand what do the agents have access to and how they use it.
- I am not sure to see what "real life" situation the model should represent and the simulations (Section 5) do not help me: here you consider a flow of images that are all of the same types and the goal is to recover the input class of the source?

---

> ### Author Response · Authors · 2023-05-17
> **Response to Reviewer rDXw**
>
> We appreciate the time and effort you have taken to review our manuscript and provide your valuable feedback. We have considered your comments and suggestions in detail. Our responses are presented below.
>
> ### Weaknesses:
>
> **Reviewer**: problem formulation is not clear… what do the agents have access to and how do they use it.
>
> **Response**: Thank you for this valuable feedback. We have revised section 2 to improve clarity. Each agent knows the possible classes $\Theta$ in the network but only has access to a local classifier $f_i$ that can distinguish only a subset of classes, denoted $\Theta_i \subseteq \Theta$. At each time step $t$, each agent $i$ receives a private signal $x_{i,t}$ sampled from the same class $\theta^*$ and obtains a posterior probability $p(\theta|x_{i,t}), \forall \theta \in \Theta_i$ from its classifier. The agents then use the proposed local and global update rules to collaboratively identify the underlying true class $\theta^*$ from the private signals.
>
> **Reviewer**: what "real life" situation the model should represent
>
> **Response**: Thank you for the suggestion. We have revised the Introduction to include more examples and motivations: “In addition to communication and computation challenges, distributed agents often possess partial knowledge and must make decisions within constraints. For example, in third-generation surveillance systems Valera & Velastin (2005) with a large number of monitoring points, camera sensors are limited by their field of view Patricio et al. (2006). Environmental and industrial monitoring involve agents with diverse sensor types collecting various data Valverde et al. (2011). Multi-agent object recognition, first introduced by Yanai & Deguchi (1998), utilizes low-cost agents identifying only a single class, to achieve wide-range recognition by increasing the number of agents and including human agents YeeWai et al. (2020). In activity recognition scenarios, various sensors, each limited in its ability to obtain full information, e.g., accelerometer, gyroscope, and magnetometer in mobile phones, are employed to classify human activities, and various vehicle sensors are utilized to classify vehicles Smith et al. (2017).”
>
> Additionally, in Section 5.2., we demonstrate the advantages of the proposed algorithm with an environment survey (object recognition) application in mind. Each agent can identify a subset of classes through a pre-trained classifier and is given a stream of images sampled from the same class. The objective is to correctly identify the object class through weak and partially informative classifiers.
>
> ### Requested changes:
> **Reviewer**: section 2 should be rewritten. a. What is $p_i(\theta)$ and why should this be known to agent i? b. What do the agents know exactly?
>
> **Response**: We have revised Section 2 as suggested to improve clarity.
>
> a.	$p_i(\theta)$ is the probability of encountering a class, dependent on domain knowledge, the environment where the agent is deployed, etc. If this is known to an agent, the convergence will benefit from it; if not, it is assumed that the probability of encountering each class is uniform across all possible classes and computed by $1/|\Theta|$.
>
> b.	As in the response above, each agent knows the number of possible classes $|\Theta|$ and has a pre-trained classifier $f_i$ that distinguishes a subset of classes $\Theta_i$ in the world.
>
> **Reviewer**: Section 3 starts with a long discussion about (Equations (3)-(6)) before ending with an algorithm that seems much simpler. Where is the catch?
>
> **Response**: The discussion demonstrates the steps we took leading to the proposed local update rule. It shows the correctness of our local update. We revised the beginning of Section 3.1 “In this subsection, we first present the derivation leading to the local updates and subsequently discuss modifications that account for the partial information available to each agent” to clarify this.
>
> **Reviewer**: Some discussion about why this problem is interesting
>
> **Response**: Thank you for this great suggestion. We have modified the introduction according to your suggestion. Please see the response above on providing real-life applications.

---

> > ### Author Response · Authors · 2023-05-17
> > **Response to Reviewer rDXw (Continued)**
> >
> > **Reviewer**: A better connection with related work would be appreciated.
> >
> > **Response**: This is a great suggestion. Our work is different from the bulk of existing non-Bayesian social learning works where each agent has **complete and precise** knowledge of the signal structure, i.e., the likelihood functions of the underlying generative mechanism and sensor characteristics for all classes and agents. As illustrated in Fig. 1, while prior works directly use outputs from sensors with known signal structures, our work directly leverages the posterior probabilities generated by any classifiers, enabling a wider scope of applications.
> >
> >  We have revised the comparison with Mitra et al. 2021 and other non-Bayesian social learning works (see the end of Sec 3.2):
> > “Although we adopt the global min update rule from Mitra et al. (2021), our problem formulation and theoretical performance guarantees differ significantly. Firstly, we bridge the gap between distributed classification and non-Bayesian social learning. Unlike prior works that require **precise** knowledge of the underlying generative mechanism and signal structure, we leverage posterior probabilities from any classifier. As illustrated in Fig. 1, while prior work directly uses outputs from sensors with known signal structures, our work directly leverages the outputs generated by classifiers. Our work facilitates the application of both generative and discriminative classifiers, enhancing model flexibility and applicability. Secondly, our approach incorporates a unique input structure, where agents are partially informative, i.e., provide information for a subset of classes. In contrast, traditional social learning requires each agent to have **complete** likelihood functions for all possible classes, i.e., $\{p_i(\cdot|\theta_q)\}_{q=1}^m$. By adopting the proposed approach, we eliminate the extensive modeling efforts required to characterize sensor and signal structures, consequently reducing overall modeling and training requirements. Thirdly, we identify, define, and quantify the roles of support agents, a crucial aspect overlooked in prior work. These agents, while not able to directly distinguish the true class, contribute to rejecting false classes. In particular, we show that the performance is influenced by both source and support agents in the network, and offer performance guarantees that demonstrate a strict improvement over Mitra et al. (2021).”
> >
> > In addition, we added a new example with a simple setup to demonstrate the effectiveness of our proposed approach compared to (Mitra et al. 2021). In Sec 5.1, two agents with simple classifiers were given data sampled from the same class. It was shown that our proposed approach outperforms common approaches such as averaging and taking maximum scores. Moreover, obtaining the required fully informative likelihood function, required by (Mitra et al. 2021) and most non-Bayesian social learning work, even for this simple set up is infeasible. We include the full discussion of the comparison here:
> > “Achieving the performance demonstrated is impossible with methods proposed by Mitra et al. (2021) and most non-Bayesian social learning works. Prior works directly use outputs from sensors (see Fig. 1) with known signal structures and are based on the critical assumption that the local signal structures (characterized by likelihood functions) for all classes are known precisely. As depicted in Fig. 2, obtaining the likelihood functions for all classes of each agent is infeasible with only observation data at deployment. Even extensive training on the likelihood functions, such as Hare et al. (2021), requires knowledge of the family of distributions and introduces additional uncertainties. On the other hand, our proposed method allows for efficient classification, even when agents are equipped with simple discriminative classifiers.
> > This signifies a substantial improvement in terms of practicality and efficiency for a broader application in real-world scenarios.”
> >
> > **Reviewer**:  I understand that there are space constraints but some element of proofs would be appreciated.
> >
> > **Response**: We acknowledge the importance of providing proof elements. Due to space constraints, complete proofs are provided in the supplementary material for in-depth understanding.

---

> > > ### Comment · Reviewer_rDXw · 2023-05-17
> > > **It might be me but I found the paper still unclear**
> > >
> > > To me, the changes to section 2 are very minimal and I do not really understand the model. It seems to me that the model should be presented in a completely different order, starting from the existence of a single class, then the observable x, etc.  To me, the paper cannot be accepted if this is not done.
> > >
> > > Also, I am not convinced by the examples of applications that are given by the authors. To me, the paper is not about classification but about identifying by which source is generated a stream of data.
> > >
> > > To conclude, to me none of the issues that I raised are solved by the revision.

---

> > > > ### Author Response · Authors · 2023-05-18
> > > > **Response to Review RDXw - formulation clarity**
> > > >
> > > > We appreciate your continuous engagement and invaluable suggestions that are helping us to improve the presentation and clarity of our paper.
> > > >
> > > > In response to your suggestions, we have revised Section 2.1, rearranging the presentation to follow the order of classes, observations, agents, and communication topology.
> > > >
> > > > As to the discussion on classification versus identification of data sources, we maintain our view that our problem is a classification task. Each input vector $x_{i,t}$ in our model is eventually assigned to class $\theta^*$ from one of the $m$ possible classes $\Theta =$ {$\theta_1,\ldots,\theta_m$}, since as $t\to\infty$, $\mu_{i,t}(\theta^*)\to1, \forall i \in \mathcal{V}$ and $\mu_{i,t}(\theta)\to 0, \forall i \in \mathcal{V}$ and $\forall \theta \in \Theta\setminus${$\theta^*$}. This is consistent with the definition of classification as found in Christopher Bishop's "Pattern Recognition and Machine Learning" (2006), page 179: “The goal in classification is to take an input vector $x$ and to assign it to one of $K$ discrete classes $C_k$ where $k = 1,...,K$”. Nevertheless, we respect the reviewer's perspective and, if the term "classification" is not seen as suitable, we are open to employing a different term that more accurately reflects our work.

---

### Review · Reviewer_mRJk · 2023-03-27

**Summary Of Contributions:**

This paper studies a distributed classification problem based on a network of partially informative agents, i.e., agents who have only a local classifier that distinguish between a subset of the entire classes. An iterative and distributed algorithm is developed that relies on belief updates using the posterior probabilities, local signals and belief information shared from neighbors. The paper shows that the algorithm converges to the true belief asymptotically, and at an exponential rate, as supported by numerical evidences.

**Audience:**

Yes

**Broader Impact Concerns:**

There are no broader impact concerns.

**Claims And Evidence:**

Yes

**Requested Changes:**

- Please state clearly what are the main differences between the algorithm proposed in this work and (Mitra et al., 2021).
- One of the main contributions of the paper seems to lie with the use of partially informative agent, and it seems to be the reason that leads to the different update rule in (7), (8) from Mitra et al. (2021). Can the paper elaborate more about the technical challenges it overcomes in showing the convergence/correctness of the proposed algorithm?
- Is the condition of uniform initial belief in Assumption 4.1 necessary? Given that the agents are partially informative, it seems a bit strange to require that the agents know what is the cardinality $|\Theta|$.
- Is it possible to give an interpretation for the asymptotic rate proven in (11)? E.g., what is the meaning of $R_{v_\theta}$?
- For the experiments in Sec 5.2, is it done with $|\Theta_i| = 10$? i.e., the local agent is fully informative. From Fig. 4 it seems that the the belief for all classes are updated in the algorithm.

**Strengths And Weaknesses:**

Strengths:
+ The problem/scenario tackled is reasonable. I believe that it describes a common scenario of partially informative agents in practical ML, and the paper is supported by a set of reasonable experiments.
+ The convergence guarantees for the proposed algorithm is well appreciated and most of the assumptions made are reasonable.

Weaknesses:
+ The paper can be difficult to read in some parts, e.g., in the introduction, it describes to a long length about the high level idea of the proposed algorithm before introducing it (p.2). It maybe better to shorten the discussion and describe the algorithm more clearly.
+ The paper mentioned multiple times that the proposed algorithm operates on the posterior probability directly (instead of the likelihood function as in (Mitra et al., 2021)). If I understand it correctly, it seems that this difference originates from the partial informative agent assumption and this brings about a small change to the algorithm in (7), (8). The discussion/comparison with (Mitra et al., 2021) is not very clear.
+ It seems that compared to (Mitra et al., 2021), the main innovation of the paper is to use a new update rule in (7), (8). In this regard, the contribution seems to be incremental as the main result proven also follows a similar favor as (Mitra et al., 2021). Together with the lack of comparison to (Mitra et al., 2021), it is hard for the reviewer to judge the paper's contribution at the moment.
+ On the side note, in light of the above, the reviewer is a little bit skeptical about the claim that the problem setup is "fundamentally different from the existing work". This seems to be slightly over-claimed as the algorithm/setup which uses Bayes rule for updating the belief has followed largely from (Mitra et al, 2021) as well as other prior works.

---

> ### Author Response · Authors · 2023-05-17
> **Response to Reviewer mRJk - 1**
>
> We are grateful for your insightful and comprehensive review of our manuscript. Your time and effort in providing constructive feedback is highly appreciated. Below, we have addressed each comment in detail:
>
> ### Weaknesses
>
> **Reviewer**: shorten the discussion and describe the algorithm more clearly
>
> **Response**: Thank you for your suggestion. We have revised the introduction and updated Fig. 1 to improve clarity.
>
> **Reviewer**: the proposed algorithm operates on the posterior probability directly and this difference originates from the partial informative agent assumption.
>
> **Response**: Unlike prior works that require **exact and complete** knowledge of the signal structure, i.e., the likelihood functions of the underlying generative mechanism and sensor characteristics for all classes and agents, our work employs machine learning models to broaden applicability.
> As shown in Fig. 1, while prior work directly uses outputs from sensors with known signal structures, our work directly leverages the posterior probabilities generated by any classifiers, enabling a wider scope of applications.
> Our proposed local update improves inference results, even with **fully informative weak classifiers in a centralized scenario**, as shown in Sec 5.2.1. Moreover, in Sec 5.1, we included a new example where our proposed algorithm is effective but acquiring fully informative and precise likelihood functions is impossible. Finally, prior works did not consider the partial information or the roles of support agents. We list the key differences in the response below.

---

> > ### Author Response · Authors · 2023-05-17
> > **Response to Reviewer mRJk - 2**
> >
> > **Reviewer**:  lack of comparison to Mitra et al., 2021
> >
> > **Response**: We appreciate this valuable feedback and have revised the comparison in greater detail (see Sec 3.2): “Although we adopt the global min update rule from Mitra et al. (2021), our problem formulation and theoretical performance guarantees differ significantly. Firstly, we bridge the gap between distributed classification and non-Bayesian social learning. Unlike prior works that require **precise** knowledge of the underlying generative mechanism and signal structure, we leverage posterior probabilities from any classifier. As illustrated in Fig. 1, while prior work directly uses outputs from sensors with known signal structures, our work directly leverages the outputs generated by classifiers. Our work facilitates the application of both generative and discriminative classifiers, enhancing model flexibility and applicability. Secondly, our approach incorporates a unique input structure, where agents are partially informative, i.e., provide information for a subset of classes. In contrast, traditional social learning requires each agent to have **complete** likelihood functions for all possible classes, i.e., $\{p_i(\cdot|\theta_q)\}_{q=1}^m$. By adopting the proposed approach, we eliminate the extensive modeling efforts required to characterize sensor and signal structures, consequently reducing overall modeling and training requirements. Thirdly, we identify, define, and quantify the roles of support agents, a crucial aspect overlooked in prior work. These agents, while not able to directly distinguish the true class, contribute to rejecting false classes. In particular, we show that the performance is influenced by both source and support agents in the network, and offer performance guarantees that demonstrate a strict improvement over Mitra et al. (2021).”
> >
> > In addition, we added a new example with a simple setup to demonstrate the effectiveness of our proposed approach compared to (Mitra et al. 2021) and other non-Bayesian social learning works. In Sec 5.1, two agents with simple classifiers were given data sampled from the same class. It was shown that our proposed approach outperforms common approaches such as averaging and taking maximum scores. Moreover, obtaining the required fully informative likelihood function, required by (Mitra et al. 2021) and most non-Bayesian social learning work, even for this simple set up is infeasible. We include the full discussion of the comparison here:
> > “Achieving the performance demonstrated is impossible with methods proposed by Mitra et al. (2021) and most non-Bayesian social learning works. Prior works directly use outputs from sensors (see Fig. 1) with known signal structures and are based on the critical assumption that the local signal structures (characterized by likelihood functions) for all classes are known precisely. As depicted in Fig. 2, obtaining the likelihood functions for all classes of each agent is infeasible with only observation data at deployment. Even extensive training on the likelihood functions, such as Hare et al. (2021), requires knowledge of the family of distributions and introduces additional uncertainties. On the other hand, our proposed method allows for efficient classification, even when agents are equipped with simple discriminative classifiers. This signifies a substantial improvement in terms of practicality and efficiency for a broader application in real-world scenarios.”

---

> > > ### Author Response · Authors · 2023-05-17
> > > **Response to Reviewer mRJk - 3**
> > >
> > > **Reviewer**: problem setup is "fundamentally different from the existing work"
> > >
> > > **Response**: We highlight that our problem setup is fundamentally different from the bulk of existing non-Bayesian social learning works where each agent has **complete and precise** knowledge of the likelihood functions of all classes. As illustrated in Fig. 1, while prior work directly uses outputs from sensors with known signal structures, our work directly leverages the posterior probabilities generated by any classifiers.
> > >
> > > In Related Literature, we state: “Prior works assume each agent has exact and complete knowledge of the local likelihood functions of all classes, known as the private signal structure (e.g., Jadbabaie et al. (2012); Nedić et al. (2017); Lalitha et al. (2018); Mitra et al. (2021)). Other works attempt to estimate these likelihood functions (e.g., Hare et al. (2021)). These assumptions necessitate domain knowledge of the generative mechanism and all sensor characteristics and can cause model misspecifications or introduce additional uncertainties. Distributed non-Bayesian learning then considers scenarios where subsets of classes are observationally equivalent (i.e., the conditional likelihood distributions of given signals are identical) at each agent, which necessitates cooperative decision-making to identify the true class.
> > >
> > > Our problem setup is fundamentally different from the existing work in distributed non-Bayesian social learning. Instead of relying on complete knowledge of the likelihood function and signal structure, our work leverages advancements in machine learning and directly utilizes the posterior probability provided by classifiers to identify the true class. These classifiers can include both discriminative models (such as random forests and neural networks, which often outperform generative models) and generative models that utilize likelihood functions (such as Naive Bayes). We assume that each agent is partially informative, i.e., each can provide information and distinguish between only a subset of classes while providing no information on the others.”
> > >
> > > ### Requested changes:
> > > Thank you again for your time providing the detailed comments. Please find our response to the requested changes below.
> > >
> > > **Reviewer**: main differences between the algorithm proposed in this work and (Mitra et al., 2021)
> > >
> > > **Response**: Please see the above response.
> > >
> > > **Reviewer**: elaborate more about the technical challenges
> > >
> > > **Response**: The technical challenges lie in addressing partial agent information, integrating posterior probabilities, quantifying the roles of support agents, and enhancing the asymptotic convergence rate.
> > > We provide local updates to not only directly integrate posterior probability but also to address the partial information of each agent. Additionally, one key observation we identified is the roles of the support agents. Although they cannot directly distinguish the true class, they contribute to the rejection of false classes. We theoretically defined the roles of support agents and their confusion scores to quantify the power of rejection.  With the key observation and technical contributions, we strictly improved the asymptotic convergence rate of the true class (Theorem 4.5).
> > >
> > > **Reviewer**: Is the condition of uniform initial belief in Assumption 4.1 necessary? Given that the agents are partially informative, it seems a bit strange to require that the agents know what is the cardinality.
> > >
> > > **Response**: Thank you for pointing it out. The uniform belief assumption in 4.1 is necessary. While initializing $\pi_{i,0}(\theta) \leq 1/|\Theta|$ is sufficient to guarantee a lower bound on the local belief of the true class, i.e., $\pi_{i,\infty}(\theta^*) \geq \pi_{i,0}(\theta^*) > 0$ (see Equation 13 in the appendix), the uniform initialization ensures a valid probability belief vector, i.e. $\sum_{\theta\in\Theta} \pi_{i,t}(\theta) = 1$.
> > > Although each agent is assumed to know all the possible classes $\Theta$, it cannot distinguish or produce any information on $\Theta \setminus \Theta_i$. The assumption of uniform belief can be easily satisfied through unique class label indexing, e.g. [1,...,m], and can be achieved at deployment or through agent communication.

---

> > > > ### Author Response · Authors · 2023-05-17
> > > > **Response to Reviewer mRJk - 4**
> > > >
> > > > **Reviewer**:  give an interpretation for the asymptotic rate proven in (11).
> > > >
> > > > **Response**: Agent $v_\theta$ is the best agent who can reject a false class $\theta$ with the highest $R_{v_\theta}$ (discriminative or confusion score of this agent).
> > > >
> > > > For example, given a clear observation of a “cat”, a support agent limited to distinguishing between vehicles cannot label it as “cat”, but contribute to global identification by informing its neighbors that the observation is unlikely to be “plane”.  If a source agent who can distinguish between “cat” and “plane” receives a blurry “cat” image, it is uncertain of the true class. Through elimination by both source and support agents, the false class is rejected. The speed of the rejection is quantified by the support (or source) agent with the highest confusion (or discriminative) score.
> > > >
> > > > Please see the discussion following (11) for the interpretation: “With probability 1, each agent will be able to reject any false class $\theta$ exponentially fast, with a rate that is eventually lower-bounded by the best with the highest performance score $R_{v_\theta}$, either the best source agent $i\in\mathcal{S}(\theta^*,\theta)$ and its discriminative score $D_i(\theta^*,\theta)$ or the best support agent $i \in \mathcal{U}^{\theta^*}(\theta)$ and its confusion score $\max_{\hat{\theta}\in\Theta_i} D_i^{\theta^*}(\hat{\theta},\theta)$ in the network. This lower bound is a strict improvement over Mitra et al. (2021), as both the source agent and support agent can contribute to the prediction convergence.”
> > > >
> > > > **Reviewer**:  Sec 5.2 … the local agent is fully informative.
> > > >
> > > > **Response**: The experiment in Sec 5.2.1 assumes the agent is fully informative to demonstrate the advantages of our proposed local update rule under the fully informed and centralized setting. Even in this fully informative and centralized scenario, common approaches such as averaging and taking the maximum fail. In Sec 5.2.2, the agents are partially informative and fully distributed.

---

> > > > > ### Comment · Reviewer_mRJk · 2023-05-30
> > > > > **Reply to authors**
> > > > >
> > > > > The reviewer appreciates the authors' efforts in revising the paper and providing a detailed response. While there are some improvement over the first version with the paper organisation, I still find the paper confusing to read, especially for the proposed setup of partial informative model.
> > > > >
> > > > > While some motivating examples have been added in the revision (in Introduction), they only pertain to the partial informative setting in a high level way. It is not clear how they fit into the particular Definition 2.1, e.g., why would an agent with cameras with partial capability get only a partial set of hypothesis class (while assuming that the agent knows that some other classes exist) instead of the full hypothesis class with low confidence. Also, I would like to see application examples that are of broader interest.
> > > > >
> > > > > Additionally, regarding the comparison to Mitra et al. 2021, the authors should consider including a concrete (i.e., mathematical) discussion to highlight the exact difference.
> > > > >
> > > > > Overall, I find the contribution to be lacking in motivation and clarity for the TMLR audience.

---

### Review · Reviewer_QNnm · 2023-05-05

**Summary Of Contributions:**

This paper considers the problem of classification in a decentralized network of partially informative agents. Here "partially informative" means that each agent can only distinguish among a subset of all possible classes. A global identifiability assumption ensures that, if the agents were to cooperate, then in principle they should be able to collectively discern which is the true class. Rather than explicitly assuming that the likelihood function associated with each class is known, the setup assumes each agent has a pre-trained classifier which can be applied locally to make predictions about the subset of classes it can distinguish. The paper proposes a distributed approach where, at each time step, every agent obtains a new (statistically independent) observation of the environment, updates its local beliefs, and then communicates with neighboring agents to "fuse" predictions.

The claimed contributions are:
1. Proposing a local update rule for this problem setup that can be applied with any local classifier.
2. Identifying conditions under which the method is guaranteed to converge asymptotically.
3. Characterizing the rate of convergence as a function of properties of the local classifiers.
4. Providing simulations to validate the convergence of the proposed approach.


**Audience:**

No

**Broader Impact Concerns:**

(Updated after author responses and revised version were posted)
The introduction mentions several motivating applications for this work related to surveillance and monitoring systems. Given this, I would expect to see a discussion of the broader impacts and ethical implications of this work.

**Claims And Evidence:**

Yes

**Requested Changes:**

To consider this paper acceptable, I would expect to see major changes that address all the weaknesses mentioned above. In particular, the motivation for the proposed setting must be made much stronger.

**Strengths And Weaknesses:**

## Strengths
1. The paper is generally well-written and easy to follow.
2. The theoretical claims appear to be correct/valid.

## Weaknesses
1. The problem formulation that this paper focuses on is not currently well-motivated. I would need to see a much stronger motivation to be convinced that at least some individuals in TMLR's audience may be interested in knowing the findings of this paper. In particular, can you provide a concrete scenario that matches the assumptions imposed in this work? (Agents cooperatively solving a classification problem, where any individual agent is only able to make classifications among a subset of all classes)

2. A secondary limitation is that the results are not particularly surprising. The assumptions, especially about Global Identifiability, ensure that as long as agents combine the outputs of their local classifiers in a sensible way, they should converge to the true class asymptotically. I fully acknowledge that proving this still requires effort to make sure things work out, but compounded with the weak motivation for the problem formulation, this further reduces my confidence that there will be interest in the TMLR audience for this work.

3. The assumption about connectivity should be stated more explicitly. Currently it's easy to miss; I had to scan several times before I found it in the sentence that spans pages 6 and 7. It would be clearer to state this explicitly in Section 2.1. (The wording just before Theorem 4.4 currently says "recall that..." but I couldn't find any mention of it before that point.) Also, if the core assumption is only that every agent has a path connecting them to source agents for every class, it would be much more interesting to state the assumption in this more general way. It could also be nice to add a comment in Section 6 discussing how the result might be generalized to time-varying topologies satisfying an appropriate related condition.

Additional minor weaknesses/comments:
* The introduction mentions robotic networks while discussing computational constraints. Most robots require substantially more energy to power their actuators than is required for computation, so this statement is not convincing. Moreover, low-power mobile accelerators are becoming much more widely available.
* In Section 2, why is $\mathcal{X}$ a discrete set? Most sensors typically produce readings in a continuous interval (i.e., floats).
* In Definition 2.3 and Assumption 2.4, the notation $\Theta_i$ is overloaded --- it was already defined in Section 2.2 as the subset of classes distinguishable by agent $i$.
* In the simulations, does every agent observe the entire image? What changes in observations at a given agent over time?
* In Theorem 4.2, claim (i), clarify that this is for $\theta \ne \theta^*$.

---

> ### Author Response · Authors · 2023-05-17
> **Response to Reviewer QNnm**
>
> We deeply appreciate the time and effort taken to provide valuable feedback on our manuscript. We have carefully addressed each comment and we feel the changes have markedly improved the manuscript:
>
> ### Weakness:
>
> **Reviewer**:  The problem formulation that this paper focuses on is not currently well-motivated. Can you provide a concrete scenario that matches the assumptions imposed in this work?
>
> **Response**: This is a great suggestion. We have updated the introduction to include several examples where agents collaboratively achieve classification with partial information: “In addition to communication and computation challenges, distributed agents often possess partial knowledge and must make decisions within constraints. For example, in third-generation surveillance systems Valera & Velastin (2005) with a large number of monitoring points, camera sensors are limited by their field of view Patricio et al. (2006). Environmental and industrial monitoring involve agents with diverse sensor types collecting various data Valverde et al. (2011). Multi-agent object recognition, first introduced by Yanai & Deguchi (1998), utilizes low-cost agents identifying only a single class, to achieve wide-range recognition by increasing the number of agents and including human agents YeeWai et al. (2020). In activity recognition scenarios, various sensors, each limited in its ability to obtain full information, e.g., accelerometer, gyroscope, and magnetometer in mobile phones, are employed to classify human activities, and various vehicle sensors are utilized to classify vehicles Smith et al. (2017).”
>
> **Reviewer**: The assumptions, especially about Global Identifiability, ensure that as long as agents combine the outputs of their local classifiers in a sensible way, they should converge to the true class asymptotically.
>
> **Response**: Thank you for pointing out this was unclear. Certainly, combining things in a "sensible" way would lead to correct classification, but the contribution of our work is proving and validating that the proposed update with the “min” rule is better than other common approaches such as averaging or maximum. It is not obvious a priori that taking the minimum over the neighbors' beliefs is a better approach than conventional methods. This paper shows that it is indeed better, and we provide theoretical analysis and simulations showing how support agents can contribute to correct classification with min-rule. The ability of support agents to help in global classification (even though support agents cannot distinguish between the relevant classes on their own), in particular, has not been considered in prior work.
>
> Additionally, we point out the global identifiability assumption is standard in non-Bayesian social learning (Jadbabaie et al. (2012); Mitra et al. (2021)). It is necessary even in a fully informative and centralized scenario under the assumption of independent signals. Consider a fully informative and centralized scenario, where an agent wants to distinguish between $\theta_p$ and $\theta_q$. Global identifiability fails, i.e., $D(\theta_p,\theta_q) = D(\theta_q,\theta_p) = 0$, if and only if $p(\theta_p|x)p(\theta_p) = p(\theta_q|x)p(\theta_q)$ for all $x\in\mathcal{X}$ from Equation 1. When the global identifiability assumption fails, there is no hope of identifying the true class as the observation data produces identical beliefs.
>
> **Reviewer**: The assumption about connectivity should be stated more explicitly. Every agent has a path connecting them to source agents for every class, it would be much more interesting to state the assumption in this more general way… Add a comment in Section 6 discussing how the result might be generalized to time-varying topologies.
>
> **Response**: This is an excellent suggestion. We have updated Section 2.1 to explicitly state that the communication graph is connected and time-invariant.
>
> We included a discussion towards the end of Sec 3.2 of the global update rule to address the comments on reachability and time-varying connectivity and we include it here: “For simplicity of analysis, we assume a connected communication network in this work. However, as demonstrated by Mitra et al. (2021), network-wide inference can be achieved as long as the source agent is reachable by other agents in the network. Furthermore, in the case of time-varying communication graphs, the algorithm remains effective when the union of the communication graph is jointly strongly connected.  A similar analysis can be used to show that the update rule in our paper (leveraging posterior distributions directly instead of likelihoods as in Mitra et al. (2021)) will also work in time-varying networks (as long as the unions of the networks over bounded intervals of time are connected).”

---

> > ### Author Response · Authors · 2023-05-17
> > **Response to Reviewer QNnm (Continued)**
> >
> > ### Minor weaknesses/Comments:
> > Thank you for carefully reviewing our paper and suggesting comments. Please find our responses below.
> >
> > **Reviewer**: Most robots require substantially more energy to power their actuators than is required for computation, so this statement is not convincing. Moreover, low-power mobile accelerators are becoming much more widely available.
> >
> > **Response**: We agree with your comment on robotic systems. Our focus on computational constraints relates to deploying large models on distributed agents. The computation is a bottleneck in real-time and on-device inference scenarios and motivates the development of lightweight models for mobile and embedded systems, such as MobileNet (Howard et al., 2019) and YOLO (Bochkovskiy et al., 2020).
> >
> > While advancements in low-power mobile accelerators are noteworthy, their deployment with large-scale real-time applications remains challenging (Howard et al., 2019; Bochkovskiy et al., 2020). Our research focuses on developing simple and model-agnostic update rules that can effectively operate on both heavyweight and lightweight models, directly utilizing posterior probabilities.
> >
> > **Reviewer**:  why is $\chi$ a discrete set? Most sensors typically produce readings in a continuous interval
> >
> > **Response**: This is a great observation. We assume the signal space $\chi$ is a finite input space for the convenience of theoretical derivation. This circumvents the issue in the theoretical analysis where the probability of receiving any particular data $x$, i.e.,  $p(x)$ or $p(x|\theta)$, in an infinite set of possible values is zero. However, in our proposed algorithm, we only used the posterior probability $p(\theta|x)$ which is assumed to be non-zero from the output of the classifier. This holds true irrespective of whether the input space is continuous or discrete, thus making our approach broadly applicable.
> >
> > While sensors can produce continuous readings, these values are often discretized due to the finite numerical precision. Data features are often discrete in applications such as text classification or image processing (see Bishop, Christopher M, 2006. "Pattern Recognition and  Machine Learning” for further insights).
> >
> > **Reviewer**: In Definition 2.3 and Assumption 2.4, the notation $i$ is overloaded.
> >
> > **Response**: Thank you for pointing out the issues with $\Theta_i$. It was included in Definition 2.3 and Assumption 2.4 to further quantify the discriminative score and define a source agent.
> >
> > **Reviewer**: In the simulations, does every agent observe the entire image? What changes in observations at a given agent over time?
> >
> > **Response**: In Sec 5.2, each agent observes the entire image. At each time step, an image is independently sampled from the entire testing set of the same class, and given to each agent. This implies that the images at a given agent could change over time, providing varying inputs to the classifier and hence influencing its output probability.
> >
> > **Reviewer**: In Theorem 4.2, claim (i), clarify that this is for $\theta \neq \theta^*$.
> >
> > **Response**: Thank you for pointing this out; we have updated Theorem 4.2.

---

> > > ### Comment · Reviewer_QNnm · 2023-05-29
> > > **Reply to authors**
> > >
> > > 1. Thank you for adding some example scenarios to the introduction. However, I still do not find these convincing. For example, one mentioned is
> > >
> > > > In activity recognition scenarios, various sensors, each limited in its ability to obtain full information, e.g., accelerometer, gyroscope, and magnetometer in mobile phones, are employed to classify human activities, and various vehicle sensors are utilized to classify vehicles Smith et al. (2017).
> > >
> > > Why does this require a decentralized solution? All sensors are on the same phone/vehicle, and so a single model has access to all of these readings.
> > >
> > > The other examples given still don't concretely motivate all of the assumptions made in the paper. For example, in an environmental or industrial monitoring system, is it even the case that a single agent may reliably classify a subset of the classes independently? It seems like it would always be preferable to fuse sensor observations directly (or sufficient statistics thereof) rather than aggregating the output of classifiers.
> > >
> > > As a separate issue, the other examples given all deal with surveillance/monitoring systems. Based on this, I would expect to see some discussion of broader impact, which is currently absent from the paper.
> > >
> > > 2. Thank you for clarifying the experimental setting in Section 5.2. This is very far removed from how an image classification system would operate, even in a decentralized setting. This connects back to the first point. It is not clear how this scenario is relevant to any (even future) system or problem setting.
> > >
> > > Overall, I maintain my position that the work is unlikely to be of interest to the TMLR audience.

---

### Decision · Action_Editors · 2023-07-19

**Recommendation:** Reject

**Comment:**

In the reviews and the discussion among the reviewers and the AE three major problems were identified:

1. Clarity: The reviewers found that the model is not clearly presented and hence the results are hard to evaluate and verify. It is not clear if the paper considers a Bayesian model or not, and if so, what information is available to each agent. $p_i(\theta|x)$ is defined as a posterior distribution representing the probability of class $\theta \in \Theta_i$ given the observed input $x$, and similarly $p_i(\theta)$ is introduced as a prior and $p_i(x|\theta)$ as the likelihood. Now the problem is the following: If these are the true probabilities (i.e., p_i = p, where p is the distribution underlying the model), then it is not clear how $p_i(\theta|x)$ is necessarily a distribution since if $p(x|\theta)=0$ for for some $x$ and all $\theta\in \Theta_i$, then $\sum_{\theta \in \theta_i} p(\theta_i|x)=0$. Furthermore, it is not clear how the agents would know exactly $p$ in a learning setting, and the potential effects of estimation errors for $p_i$ are not considered. If $p_i$ is not the true distribution, then assumptions ensuring roughly that $p_i$ is a "reasonable" estimate of $p$ (for $\theta\in\Theta_i$) are missing (which would be necessary for the main statements to hold).

2. Motivation: The reviewers found that the model is not sufficiently motivated, and the authors should present more convincing applications.

3. The incremental nature of this work and comparison to Mitra et al. (2021): The reviewers found that the improvement compared to Mitra et al. (2021) is quite incremental, and not properly explained. While a large improvement over existing literature is not a necessary requirement for TMLR, and I appreciate the simple trick of rewriting the update rule of Mitra et al. with the class posterior instead of the data likelihood, the paper should explain clearly from the beginning that this is the main result and why this simplifies the problem, that is, that the difference is if the agents need to know the distribution $p(x)$ (or $p_i(x)$), and why modeling $p(x)$ or $p(x|\theta)$ is much harder than modeling $p(\theta|x)$ (and $p(\theta)$).


A revised version of the paper addressing all the above issues may be of interest to the community.

**Audience:**

The reviewers found that the paper lacks a proper motivation and the discussed scenarios do not fit the model considered. Hence it is not clear why the model would be of interest to the readers. Having said this, if the authors can provide some more convincing applications and the clarity and correctness can be improved, the paper may become interesting to some members of the community, although the novelty compared to the work of Mitra et al. (2021) is rather limited.

**Claims And Evidence:**

The reviews and the extensive discussion among the reviewers and the AE identified some issues about the clarity and correctness of the paper, as detailed in the comment section.

**Resubmission Of Major Revision:**

The authors may consider submitting a major revision at a later time.